# Motion Shield: An Automatic Notifications System for Vehicular Communications

**DOI:** 10.3390/s22062419

**Published:** 2022-03-21

**Authors:** Petros Balios, Philotas Kyriakidis, Stelios Zimeras, Petros S. Bithas, Lambros Sarakis

**Affiliations:** 1Knowledge Society Ltd., 115 23 Athens, Greece; p.balios@knowledge.com.gr; 2Department of Statistics and Actuarial-Financial Mathematics, University of the Aegean, 811 00 Mitilini, Greece; zimste@aegean.gr; 3Department of Digital Industry Technologies, National and Kapodistrian University of Athens, 157 72 Athens, Greece; pbithas@uoa.gr (P.S.B.); lsarakis@uoa.gr (L.S.)

**Keywords:** accident prediction, automatic crash notification, decision support system, driving safety, mobile device

## Abstract

Motion Shield is an automatic crash notification system that uses a mobile phone to generate automatic alerts related to the safety of a user when the user is boarding a means of transportation. The objective of Motion Shield is to improve road safety by considering a moving vehicle’s risk, estimating the probability of an emergency, and assessing the likelihood of an accident. The system, using multiple sources of external information, the mobile phone sensors’ readings, geolocated information, weather data, and historical evidence of traffic accidents, processes a plethora of parameters in order to predict the onset of an accident and act preventively. All the collected data are forwarded into a decision support system which dynamically calculates the mobility risk and driving behavior aspects in order to proactively send personalized notifications and alerts to the user and a public safety answering point (PSAP) (112).

## 1. Introduction

The number of traffic accident fatalities and serious injuries due to the lack of early medical assistance on site is staggering. A major contributing factor attenuating the problem is the fact that, up to now, there was no technological methodology effective and robust enough to warn the car driver of circumstances leading towards the onset of an accident. Another factor is the inevitable delay between event detection and emergency medical services (EMS) notification and response. To mitigate the problem, traffic management authorities worldwide consider an in-car system approach, such as the eCall vehicle-embedded system, implemented since 2018 in the EU, in order to automatically alert EMS to provide early medical assistance on site.

Unfortunately, in-car systems have major disadvantages. They are very expensive, they can be installed in relatively new cars only, and are vehicle-dependent. Because of these inadequacies, a profound need becomes apparent: to provide protection of the individual throughout their daily activities, especially when commuting, regardless of their choice of transport. An alternative approach was soon followed.

## 2. Background

### 2.1. Literature Review

Interest around automatic crash notification systems is not something new. Efforts towards building such a system span more than half a century, based on diverse objectives and various degrees of success. We will briefly review some of the more recent advances that influenced, to a greater or lesser extent, our current work. A comparative study of the work referenced in this paper is shown in Figure 1. The system proposed in the current paper is inspired by an automatic cloud-based notification system for vehicle accidents, developed by Andreas Alamanos [1]. He proposed a method and a client-server system for transmitting automatic notifications in the event of a vehicle accident. This is achieved by monitoring sensor readings of a vehicle occupant’s mobile device to evaluate the possibility of a crash, in which case, the proposed method weeds out false positives and transmits a final alarm to an emergency service center.

Although Alamanos’s idea seems very similar to ours, we differ fundamentally in the crash management protocol. His method initiates a communication protocol during or after the end of the crash, whereas we actually predict the onset of a number of emergencies and proactively alert the driver or a public safety answering point. Fanca et al. [2] sought an effective, double-pronged approach intended to reduce the number of falls-related deaths by building a system, using mobile phones, to detect and report accidents, when they occur, and reduce the time of accident report and medical intervention to the scene of the accident. Zualkernan et al. [3] created a mobile phone application which uses an artificial intelligence-based classifier software to distinguish between various types of accidents, categorized from a collection of data, derived from simulating different kinds of collisions. The mechanism consequently notifies EMS with the accident type and car GPS location. Bahar Al-Mayouf et al. [4] established an accident management system to ensure real-time communication among vehicles, message-relay networks, and emergency medical services. The system successfully reduces the time needed to dispatch an ambulance to an accident scene, using a multihop optimal forwarding algorithm. Khan et al. [5] developed an Android application to monitor the sensors of a smartphone to detect vehicular accidents, notify EMS with real-time GPS information, and minimize the response time of emergency services. Their scope is to also address all possible kinds of emergencies with this system. Bhatti et al. [6] similarly developed an Android application to read sensor readings for speed, g-forces, pressure, sound, and location in order to detect road accidents and alert the nearest hospital. The proposed approach was developed to be easily deployable in legacy vehicles.

Again, Motion Shield addresses potential emergencies well before they arrive and acts accordingly, whereas [2,3,4,5,6] always alert EMS after the accident, without proposing any surefire way to safeguard the mobile phone from being destroyed during the accident. Alvi et al. [7] presented a critical analysis of major existing technologies associated with accident prediction, detection, and prevention, such as vehicular ad hoc networks, GPS/GSM-based systems, machine learning techniques, etc. Their comprehensive study helped clarify the objectives of the work outlined in this paper. The crucial concept of automatic collision notification availability and emergency response times following vehicle collision was studied by Russell L. Griffin et al. [8]. Using data from the 2017 Crash Investigation Sampling System, they assessed whether in-vehicle automatic collision notification systems played a significant role in faster emergency medical services notification and transfer to a medical center from the site of an accident. To ascertain the potential of reduced timeframes, they used a Cox proportional hazards model. Therefore, ref. [9] analyzed existing technologies and ref. [10] assessed a methodology, employing vehicle-dependent ACNs, to reduce EMS response times. Both theoretical studies offer valuable insight into Motion Shield’s applied approach.

### 2.2. Terminology and Definitions

Key terms and comprehensive definitions used throughout this work are collectively presented in Table 1 in order to delineate the major concepts, illuminate the operational logic of Motion Shield, and help make them easily accessible.

## 3. Problem Description and Paper’s Contributions

### 3.1. Problem Description

Software-based ACN systems, exploiting the versatility of mobile devices, were eventually developed ([2,3,4,5,6], as commercial apps PODIS, Collision Call, SOSmart, DriveSense, Blinkapp, ROADSAVE, SafeBeep). However, both hardware- and software-based systems have a major caveat: they send notifications after the accident. This creates two significant gaps in end user protection:The system does not know the location of the user and therefore cannot give them, proactively, any notice of possible dangers they will encounter (extremely dangerous conditions in the road segment they are driving on, immobilized cars they will encounter shortly in the direction they are headed, etc.).If an end user of the system is involved in an accident and the automatic accident notification system malfunctions for any reason (lack of network coverage, the device is destroyed before sending the signal, etc.), no one will be informed of the interruption of communication between the mobile device and the system, nor will they know the area that the end user was in before the communication was severed.

In order to address these weak points in driving safety, we need to:Develop a cost-effective, cloud-based, life-threatening event-detecting, decision support system.Most commercial systems currently in the market are not cost-effective: they require steep subscription fees, mainly due to hardware-embedded installations and maintenance costs. Mobile systems, on the other hand, fail to secure a loyal user base, due to unsuccessfully addressing, if at all, event detection and proper emergency protocol management.Resolve cases where no signal is transmitted from ACN systems, when network coverage is spotty or non-existent, or the mobile device is destroyed before it transmits the proper signal, which causes all current crash notification technologies to fail. No system, to our knowledge, hardware- or software-based, manages to effectively circumvent the thorny issue of communication loss with a GSM network. That leads to rendering any sophisticated installations powerless in the face of a potential accident.Estimate the probability of becoming involved in an accident, in case the signaling device is incapable of sending, for any reason, the expected notification. That is directly related to the previous objective and still left unmitigated by current systems.Provide personalized notifications for risks the end user might face down the road. Few mobile applications take a proactive stance towards alerting the user of an imminent danger. All react to the emergency, albeit with dubious success, in case a catastrophic system failure prevents communication between the mobile device and EMS.

### 3.2. Proposed System’s Major Features

To mitigate the aforementioned issues, we propose a new, software-based automatic crash notification system which utilizes the versatility of mobile devices and goes beyond protecting the user inside the confines of a moving vehicle only, but also becomes a personal guard in all user activities such as commuting, biking, or extreme sports. When a vehicle is moving, the system considers, in real time, all the parameters that affect driving safety, dynamically calculating the mobility risk, based on the prevailing conditions in the area of observation. For example, in road traffic circumstances, maximum speeds vary, depending on local weather conditions or various stages of disrepair of the road network (potholes, road pits, etc.). The operating principles of the system are based on a new logic:Monitor and transmit spatiotemporal and velocity data to a central information management system.Combine those with driving behavior, historical evidence of serious traffic accidents, road network topology, onboard diagnostics, and many other parameters to assess the likelihood of the user becoming involved in an accident.

This information is forwarded to a decision support system in real time, which decides when to warn users about possible hazards that could cause an accident. This approach achieves, among others,

Seamless operation, even in areas not covered by a mobile network signal. This issue is discussed in Section 4.3.3, “Reporting Services” subsection.Creation of an early warning system to raise user awareness about potentially dangerous situations they may encounter on the road. Obstacles on the road, traffic bottlenecks, accidents, and extreme weather phenomena are part and parcel of the early warning system.Assessment of the likelihood of an accident, using probabilistic and stochastic models. It will be honed and improved in future versions of the systemNotification of the public safety answering point for possible accident involvement: 112 is an emergency handling system in full operational capacity across Europe. The emergency signal transmission implemented in Motion Shield utilizes the emergency management reflexes of 112.Construction of the driving profile for improved safety and performance. Driving behavior is assessed through evaluation of parameters such as abnormal acceleration, frequent and abrupt braking, hard turns, random lane negotiation, etc. These are derived exclusively from the mobile phone sensors (linear and vertical accelerometers, gyroscope, magnetometer). They depict a very sharp image of driving habits. It is not hard to incorporate these metrics into a processing algorithm to calculate the driving risk and the likelihood of accident involvement.Roadside assistance notifications through the onboard diagnostics module. OBD-II implementation will be developed in future work.

The remainder of this article is as follows. In Section 4, the proposed system architecture is presented. In Section 5, the conclusions drawn from this experimental design are presented, and points for future work are briefly discussed.

## 4. Proposed System Architecture

The system, as shown in Figure 2, operates in three interlinked levels:The application level.The cloud level.The control level.

The interlinked levels consist of the following interconnected subsystems: The end user application installed on mobile devices regularly sends predefined content messages and emergency messages when special conditions are met; the Geographical Information System, which stores dynamic data coming from mobile devices and subscription services, as well as static data. It cooperates in tandem with the statistical processing subsystem; the central information processing system, which hosts the decision support system; the system management and the reporting services subsystems. The public safety answering point is not part of the interconnected systems, per se, but rather a significant contributor to the safety operations, since it is the crucial establishment that mobilizes medical assistance services in case of an emergency. Data from smartphone sensors, geolocation, driver notifications, and third-party information are transmitted and received via the 3G/4G/5G cellular network. The process is transparent on the air interface. The mobile device uses the 3G/4G/5G communication protocol, depending on compatibility and local availability. The mobile application is light on resources and keeps battery levels normal during operation.

### 4.1. Application Level

When the application is activated, it sends the $S_{0}$ signal, alerting the system that it is currently in standby mode. The process is described in Figure 3.

After $S_{0}$ transmission, when the speed of the MD exceeds AS, the OBD-II interface is pinged. If it is active, $S_{1}$, with which the application is placed into operating mode, is transmitted, including diagnostics from the OBD-II interface, and a new session begins. If the signal cannot be sent (lack of data transmission), it awaits *T* s and rechecks the maximum speed of the mobile device for the last *T* s. If, during this time, the maximum speed of the mobile device exceeds AS, then it sends $S_{1}$ with updated data. If the signal cannot be sent again, the same procedure is repeated every *T* s.

After sending $S_{1}$, the application is programmed to send each signal with a time deviation of *T* s from the previous signal. When $S_{n}$ cannot be sent (e.g., because of lack of mobile network coverage), the signal is sent as soon as it becomes possible. When the application operates, it transmits a data packet every *T* s. If, after $S_{n}$, the mobile device is in an area where there is no network coverage, no signal can be transmitted. In this case, the application withholds the data packet, which was next in line for transmission, updates it, and transmits the signal $S_{n\;+\;1}$ immediately after the mobile device enters an area with mobile network coverage. In this case, ($t_{n\;+\;1}-t_{n}$) is greater than *T* s. During operation, the application is in constant alert to detect emergencies (explained in Figure 4) that are either due to battery power falling below a predetermined threshold on the mobile device or to end user actions (device switch-off, data transmission deactivation), or, in extreme conditions, resulting from a possible crash or a large vertical displacement.

In these cases, the following actions are taken:If the application detects a battery power level of the mobile device below a predetermined threshold, then it immediately sends an $ES_{1}$ signal. If the signal cannot be sent, it tries again at $\frac{T}{2}$ intervals. The mobile device sends a proper signal to the DSS.If the application detects end user action to disable the data transmission, it immediately alerts the DSS by sending $ES_{2}$ before the process is completed. Then, the application enters standby mode.If the application detects the end user turning their device off, it immediately alerts the DSS by sending $ES_{3}$ before the process is completed. Then, the application enters standby mode.If the application detects a sudden major decrease in the speed of the mobile device (such as in the event of a crash), it immediately transmits $ES_{4}$ to the DSS. Then, the application continuously sends to the central system the position and the speed of the mobile device for a period of 3T+ST. After the end of that period, signals are sent every $\frac{T}{4}$ for a period of 30T, unless the central system sends a different notification. During the 30T period, the application does not enter standby mode. The central system supplies ST when communication with the application is available. In case of communication inability with the central system, ST is set to be 15 s.If the application detects a sudden major vertical displacement exceeding a predetermined threshold, it immediately sends $ES_{5}$ to the DSS. If it cannot send the signal, it does so as soon as it finds an active data transmission. The signal also conveys the coordinates of the point at which the event occurred. $ES_{5}$, along with regular $S_{n}$ signals from the same region, periodically (e.g., once a week) undergo statistical analysis from statistical processing, producing results regarding the road network surface quality, forming a separate GIS layer.

If, during the period which starts at $t_{n}$ and ends at $t_{n\;+\;1}$, the maximum values of the accelerometer and/or the gyroscope exceed predetermined thresholds, then $S_{n\;+\;1}$ includes these values in its data payload. These data also undergo statistical analysis, producing results pertinent to the road network surface quality. If the application is operating and the speed of the mobile device falls below AS for more than 3T s, then the application enters standby mode and immediately transmits the $S_{end}$ signal, indicating termination of the current session. The application supports interfacing with the OBD-II protocol to collect vehicle telemetry and provide real-time data on vehicle subsystems’ functionality (tire pressure, temperatures, hydraulics, electrical systems, etc.). The application has the ability to send notifications at the user’s request directly to a PSAP concerning medical need, safety issues, and fire risks. It also provides the ability to alert the appropriate call center for car problems. The application displays to the end user useful, personalized information coming from the system.

### 4.2. Cloud Level

At this level, subscription services are third party providers (weather data, etc.), whereas the DSS shapes the central system.

#### 4.2.1. Geographical Information System

GIS is the central database of the system that contains three data categories, obtained from subscription services or aggregated from end users:Static data, including the design of the road network, road topology, etc.Dynamic data, such as cellular network coverage, meteorological data, etc.Incoming data from the mobile device.

All the above data have a common feature: they contain geographical information, which links these heterogeneous data together.

#### 4.2.2. Statistical Processing System

Statistical results are extracted from the statistical processing system:From gyroscope recordings and mapping of dangerous road points (e.g., potholes, road pits): If a statistically significant probability shows large fluctuations with respect to the vertical axis, the central system registers a relevant GIS record for that session of the road network.From potential bottlenecks that may be encountered in the vehicle’s traveling direction during the next few minutes: If a statistically significant probability shows on different sections of the road different mobile devices that run with less than the AS, the central system registers to the GIS an indication of a possible congestion.From the creation of a model that associates the average speed of traffic in a section of the road network with visibility and weather conditions.From the creation of maps per mobile operator that reflect the poor signal and blank spots related to the network coverage: By statistical analysis of the signals not sent to the cloud, we can delineate the areas where cellular coverage is spotty or absent altogether.

#### 4.2.3. Decision Support System

The subsystem includes decision support and multi-criteria decision-making algorithms. The subsystem issues decisions on issues such as:Predicting the potential of accident involvement, accounting for the current MD movement behavior and the road section hazard risk the user will reach soon. The exported individualized result, under certain conditions, leads to the creation of an appropriate report that can be forwarded to the user to prevent an accident.The proposed maximum driving speed in a section of a roadway based on current conditions (meteorological phenomena, road condition, time and daylight, road network topology risk, etc.).Visualization of end user profiles by combining historical and current data.Notification of potentially dangerous situations encountered by the end user shortly on their route with regard to current traffic behavior, such as:(a)Traffic jams—immobilized cars they will soon encounter and are likely to create hazards on the road.(b)Crashes they will encounter in their direction.(c)Dangerous road ahead, which is characterized by a combination of factors (road topology, MD movement behavior, meteorological phenomena).The central system, upon receiving $ES_{4}$, initiates a crash management process (Figure 5), where it dynamically calculates parameters SD & ST and sends ST to the application.This step supports two different cases:(a)If, for (ST+T) s and after $ES_{4}$ transmission, the central system will not receive anything from the application, it alerts the PSAP.(b)If, for (ST+T) s and after $ES_{4}$ transmission, the central system receives signals from the application and detects motion of the mobile device from ($t_{ES}+ST$) until ($t_{ES}+ST+T$), the speed of which is larger than $\frac{AS}{2}$, then the emergency is archived. In case no movement of the MD greater than $\frac{AS}{2}$ is detected during this particular time segment, the central system queries the application and expects an answer for 2T s. If an answer is indeed received, the system archives the event or alerts the PSAP, depending on the end user’s feedback. If no answer is received during this particular time segment, the system monitors the MD’s maximum speed during this time segment and any displacement from the location $ES_{4}$ was last transmitted. If MD’s moving speed was still less than $\frac{AS}{2}$ and the displacement less than SD, the system alerts the PSAP. This way, notifications after $ES_{4}$ transmission and before PSAP alert are filtered on three different levels: the MD moving speed, the MD displacement, and the end user application confirmation (three-way accident verification).If, after time ($t_{ES}+ST+3T$), for a period of 30T, reasons for cancellation of $ES_{4}$ are in effect (retaining $u>\frac{AS}{2}$, MD displacement from the location $ES_{4}$ was last transmitted greater than SD or end user response to the special query that they are OK), the central system sends PSAP a new signal canceling the alert.In the event of communication failure with the mobile device, the DSS assesses relevant factors and decides to update the PSAP for a possible accident in the area. This is achieved by estimating the expected time the mobile device will take to send the next signal. This calculation shall be made taking into account the following:(a)The mobile network coverage map.(b)Since the mobile device has entered an area not covered by a cellular network, by calculating the expected time at which the device will exit from that area, based on driving behavior, the degree of road network topology risk, the meteorological data of the area, the distance to travel.In case of a significant delay when sending a signal in relation to the expected time of sending that signal, the decision support system considers all the factors and makes the decision on whether to update the PSAP or not.

The decision support system also includes:An expert system, XS1, of hazard assessment of the road the end user is currently on that time. To calculate the degree of risk (dynamic estimation), all parameters mentioned in the GIS are taken into account. XS1 dynamically calculates the road risk from GIS data provided by third parties. These include traffic bottlenecks, road accidents in the direction of mobility, the general condition of the road network, road works, objects on the road (roadkill, landslides, boulders), and extreme weather phenomena. When an assessment of these conditions surpasses a particular safety threshold, drivers receive notifications and warnings towards speed reduction with justified explanation.An expert system, XS2, which, acknowledging information from XS1, estimates the maximum safe speed for that road segment. XS2 monitors speed, GPS coordinates, and localized weather conditions. It also derives the road type, visibility, and brightness of day. Combining these data with XS1 and national standards for maximum safe driving speeds, XS2 produces a dynamic assessment of a real-time safe speed and notifies the driver about it.An expert system, XS3, from which the current MD movement behavior profile is derived, acknowledging information from XS2, the vehicle speed, and the maximum accelerometer values. XS3 exploits mobile device sensors to create a driving profile, based on the driver’s mobility behavior. Taking into account information from XS2, the system monitors speed and accelerometer spikes, angular velocity, braking and turning behavior, frequent acceleration and deceleration, overtaking, and smoothness in order to classify driving behavior according to specific types.An expert system, XS4, from which the likelihood of an accident involving a mobile device is derived. XS4 evaluates the likelihood of a vehicle becoming involved in an accident. It does so as a result of real-time processing from all parameters stored in GIS, the valuable contribution of expert systems XS1–XS3, and historical data for road accidents from national and international repositories. XS4 then acts on the output of information processing, alerting the driver for an imminent danger that might develop into an accident. XS4 plays a critical role in initiating the communication protocol with EMS when the likelihood of an accident breaches a safety threshold.

### 4.3. Control Level

#### 4.3.1. Evaluation Setup

Under the proposed modeling of the statistical processing system, one of the main goals is the extraction of a model, based on the road network, considering visibility and weather, under the speed of traffic (AS). Based on the system, various conditions must be introduced, combined with corresponding weights leading us to model definition. Under the final result, the decision of the individual is taken considering the road risk, based on the proposed model. The main tasks of the system are benefit maximization (reduction of risk level) and selection of the most productive action groups with higher benefit rate. To develop the model analytically, we investigate the effectiveness of the process that could be achieved considering different systems: human, environment, and infrastructure. For that reason, a proposed risk factor must be introduced as a result of a combination between various conditions (weather conditions, meteorological phenomena, road conditions, visibility, brightness, road network topology risk, speed). The overall risk assessment borrows ideas from the Failure Modes & Effects Analysis tool (FMEA, [11]), which is a best practice followed by many industries for an effective processes, designs, and services risk management, especially in areas where quantitative and qualitative elements are complimentary. A significant aspect of the tool is that failure rates are usually predicted from generic rates developed from experience over time.

The core metrics of an FMEA assessment are usually severity (of failure), probability (of occurrence), and detectability (of failure), culminating in an overall risk:(1)R=Severity×Probability×Detectability

In the case of Motion Shield, the combined conditions produce its rating risk, expressed mathematically by the general formula
(2)$$R=\prod_{i=1}^{N}S_{i}=S_{1}\cdot S_{2}\cdots S_{N}$$
where *R* is the risk factor, $S_{i}$ are the introduced conditions severity values considering various systems, and *N* is the number of conditions. The proposed model also borrows insight from the risk models based on DMRA (decision matrix risk assessment). The DMRA method is a systematic risk assessment approach that consists of measuring and classifying risk indicators and is based on a documented assessment of the possibility of an accident or failure and the consequences stemming from escalation of their severity (Henselwood & Phillips [12]; Haimes [13]; Reniers et al. [14]; Woodruff [15]). The main advantages of risk rating models can be summarized as follows:They provide immediate response.They require low-cost analysis.They are modifiable and, in a way to introduce new data.They are great tools that support distribution of decision-making resources.They allow identification and grading of alternative countermeasures.

The final results are combined into the current model, considering risk factorizations, where experts determine the risk indicators, introducing parameter weights, and calculate the overall score of the risk indicators (Suddle [16], Yoon & Choi [17]). Considering the severity of conditions ($S_{i}$) of risk factor (*R*), weights ($w_{i}$) have been calculated for each condition, transforming the general formula into
(3)$$R=\prod_{i=1}^{N}w_{i}S_{i}=w_{1}S_{1}\cdot w_{2}S_{2}\cdots w_{N}S_{N}.$$

The weights are calculated based on the variable conditions that inadvertently affect driving safety. It has been proven useful to adopt a nonlinear scoring scale, as consecutive numbers in weight values turn out to be counterintuitive because they do not allow more distinction between ratings. When there is no straightforward process to quantify qualitative parameters, we usually adopt a random scoring scale, run simulations, and improve accuracy later. Translating these values into realistic case studies, we need to develop a sophisticated testing platform, which, at the moment, is prohibitively costly. Certain conventions have been adopted in establishing a scoring scale for different conditions. For example, the lower the visibility, the higher the weighting value. In Table 2, Table 3, Table 4 and Table 5, the scoring scales per condition are highlighted.

The criteria to discriminate the levels of the risk factor, considering a proposed value (*V*), are chosen based on the thresholding process [10], where, if the risk factor (*R*) is less than the proposed values (*V*) (R<V), then no risk is taken, otherwise a risk is applied based on the following rule:(4)$$R=\left\{\begin{array}{c} \mathrm{risk},\mathrm{if}\;R\geq V\\ \mathrm{no}\mathrm{risk},\mathrm{if}\;R<V \end{array}\right.$$

Thresholding is the simplest method for categorization. In the general form, the algorithm returns a single intensity threshold that separates risk factor into two classes, TRUE (risk) and FALSE (no risk). The decision process is given in Figure 6.

#### 4.3.2. Experimental Results and System Dashboard

In this section, the simulation of the decision process has been presented, leading us to the risk factor implementation. The process could be illustrated by using two examples, considering the time transmission: 1. Fixed time t; 2. During a time period $t_{n}-t_{m}$, where n<m.


**Fixed Time t**


Considering five different conditions based on the proposing systems, human (speed), environment (weather conditions, visibility, and brightness), and infrastructure (road conditions), the general risk model is given by
(5)$$\begin{array}{cc} R=w_{1}\cdot speed\cdot & w_{2}\cdot weather\cdot w_{3}\cdot conditions\cdot w_{4}\cdot visibility\\ & \cdot w_{5}\cdot brightness\cdot w_{6}\cdot road\;conditions \end{array}$$

Under the system proposed coding, considering the scoring scales per conditions, the corresponding conditions have the following codes: speed (if 50 km) = 1.5; weather conditions (if clear sky) = 1; visibility (if 175 m) = 3.5; brightness (if night) = 1.5; and road conditions (if urban network) = 1.5. The weights for the conditions are $w_{i}=1$, for i=1,2,3,4,5 under the specific data based on the proposed system. The risk model is R=11.81; if the proposed value was V=250, then, under thresholding, process R<V, meaning there is no risk. The proposing method is presented in Figure 6.


**During a Time Period $t_{n}-t_{m}$, where n<m**


Considering a time period of 100 min, the simulation of the process, based on the conditions’ codes, is given in Figure 7.

The corresponding figure illustrates the simulation process of the risk factor (*R*) against time period (*t*) under various thresholding values (*V*). The graph represents the risk factor (*R*) simulation using the proposing conditions under the ranges of their scoring scales (Johannsen and Bille [18]; Kittler and Illingworth [19]). Based on the results, given the proposing thresholding value (*V* = 250), it is clear that, throughout the observation period, the process is safe (no risk −0% road risk), leading us to conclude that the proposing system works properly. As the thresholding value (*V*) decreases (*V* = 100), the road risk increases. Based on the simulation under the graph presentation (Figure 7), it can be seen that the risk condition R≤V(=100), in our case indicated by three position alerts (times minutes) in 100 positions (times minutes), has expected probability of (3/100) 3% of occurrence of contingency (road risk) during the time in minutes.

Figure 8 shows a screenshot of the Motion Shield system dashboard. This is where subscribers’ information is handled, incoming alerts are processed, system settings are parameterized, and statistics reports are extracted. Customization is also available, according to the corporate client’s specifications and needs. This instance does not, at the moment, include all functionalities that are being developed and tested during the software development life cycle. From a technical point of view, the dashboard backend runs on PHP 7.4 and the user interface on HTML5 and Bootstrap. The mobile app sends requests to the server, where the dashboard resides, to transmit data such as user details, GPS coordinates, and speed in JSON format, where it is then stored in a MySQL database.

#### 4.3.3. Reporting Services Subsystem

This subsystem generates reports accounting for the processing of information acquired from the application, the statistical models, and the expert systems. The process, described in Figure 9, generates reports such as:Loss of contact (flag 1) with the mobile device. If the system receives no signal while the mobile device is inside an area covered by a mobile operator, then, taking into account the latest available MD movement behavior and the delay time for signal transmission, a specific algorithm calculates the probability of the mobile device becoming involved in an accident. When the probability exceeds a predetermined value, the system reports contact loss with the mobile device to the regional PSAP.Loss of contact (flag 2) with the mobile device. If the system receives no signal while the mobile device is outside an area covered by a mobile operator, then, taking into account the latest available MD movement behavior and the delay time for signal transmission, a specific algorithm calculates the probability of the mobile device becoming involved in an accident. When the probability exceeds a predetermined value, the system reports contact loss with the mobile device to the regional PSAP.Map creation with maximum safe speeds on the road network.End user moving behavior.Verification of mobile device accident involvement.

## 5. Conclusions—Future Work

Motion Shield does not intend to be a panacea for traffic accident prevention. It is, rather, a cost-effective attempt at predicting the onset of a road traffic event, using as few resources as possible and acting on multiple levels before the event blossoms into a full-fledged emergency with dire impact on human life and infrastructure. The system intends to:Achieve a three-way method of accident verification.Dynamically estimate the maximum safe speeds on the road network based on traffic, weather, and driving patterns.Negotiate a safe outcome to a situation where the system detects sudden loss of MD signal in an area without network coverage, and calculate the probability of becoming involved in an accident inside this area.Assess the driving behavior, considering different parameters that affect driving safety.Calculate the probability of becoming involved in an accident as a function of driving behavior and hazardous road conditions.Raise awareness with personalized notifications regarding imminent accidents, traffic jams or obstacles on the road, and dangerous weather phenomena.

### Future Work

The system is under continuous development and is far from its final form. We are constantly incorporating new features, testing and evaluating new methods and processes. A major redesign is scheduled to include modules with machine learning and neural networks, which will render decision-making much more accurate and powerful. At a later stage, estimating the probability of accident involvement is a point we have to address in rigorous detail and correlate with the range of road risk evaluations. After numerous business discussions with potential strategic partners, we have established that the road risk assessment and the driving profile are of particular interest to the insurance industry, and its large customer base is an ideal bedrock for beta-testing Motion Shield on a large scale. It is large-scale testing for at least 12 months that will boost the system’s efficiency and corroborate its value.

Testing of the contributions and countermeasures proposed in Section 3.2, Proposed System’s Major Features, require significant resources and are both taxing in complexity and depth. Accident severity, detection, and management cannot be tested, primarily because of safety concerns. Vehicle destruction is also costly and is generally frowned upon. An alternative solution would be to book, for a short period of time, the crash testing platform that major automobile corporations, such as BMW and General Motors, have in their facilities. That also incurs prohibitive testing fees, traveling and accommodation costs, and involves iterative tests to obtain result consistency, and so forth. At the moment, we can only resort to simulations, which cannot accurately evaluate the proposed model and are prone to noisy interference. Comparison of systems developed by other researchers would require replicating their hardware and software implementations: vehicle-dependent systems and apparatuses fall outside the scope of the work presented in this paper, whereas software applications are either complimentary, not commercially accessible, or defunct. Motion Shield is a software-based system that aims to provide unique functionalities with as few resources and as little processing power as possible.

Last, but not least, future work will tap into the vast reservoir of useful data derived from the onboard diagnostics vehicle interface (OBD-II), which will provide us with an accurate snapshot of a vehicle’s condition. It is expected that incorporating values from the most relevant parameter IDs will achieve a higher accuracy in risk assessment and weight scaling.

## Figures and Tables

**Figure 1 sensors-22-02419-f001:**
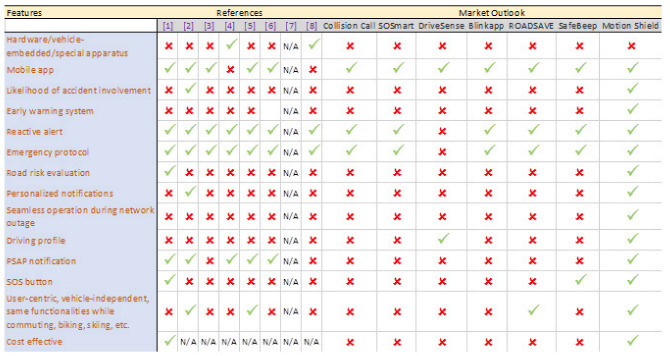
Existing systems vs. Motion Shield—a comparative study.

**Figure 2 sensors-22-02419-f002:**
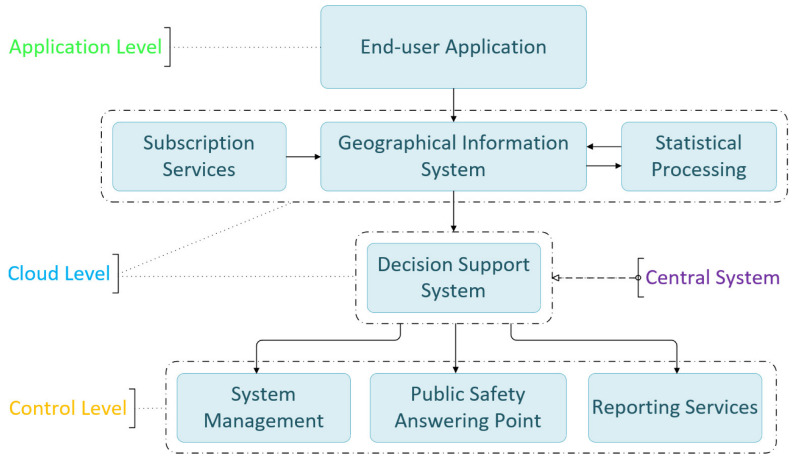
Description of the structure of the proposed system. It consists of three functioning levels: the application, the cloud, and the control level, with subsystems operating at their corresponding level.

**Figure 3 sensors-22-02419-f003:**
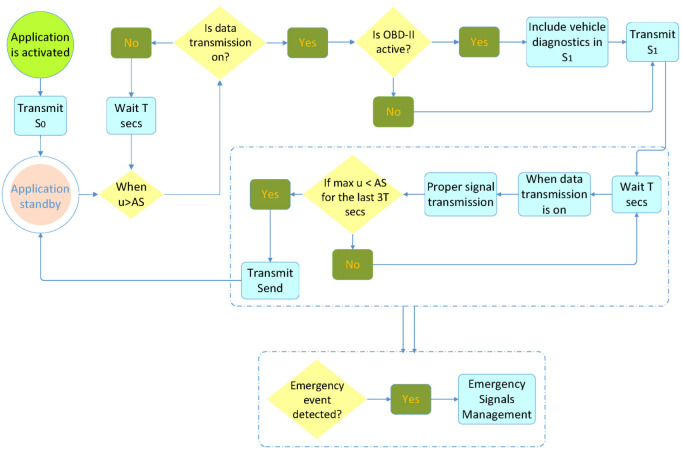
Description of how the mobile app operates in the application level.

**Figure 4 sensors-22-02419-f004:**
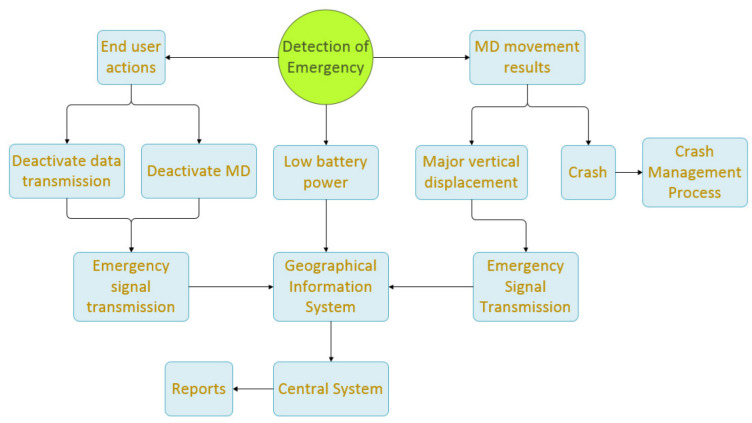
Description of what happens when the system detects an emergency (emergency signals management introduced in Figure 3).

**Figure 5 sensors-22-02419-f005:**
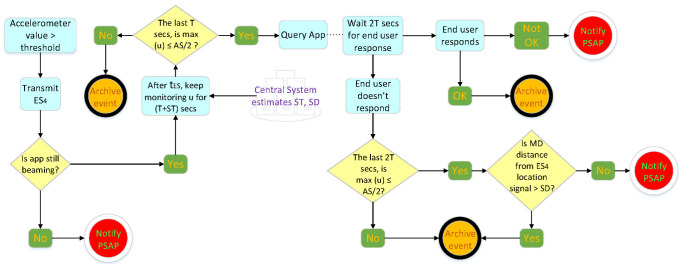
Description of the crash management process introduced in Figure 4.

**Figure 6 sensors-22-02419-f006:**
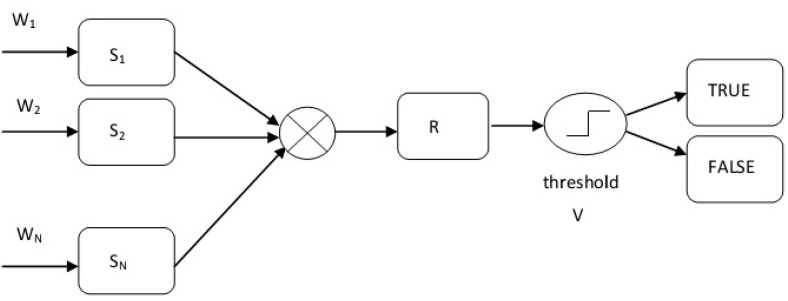
Decision process methodology.

**Figure 7 sensors-22-02419-f007:**
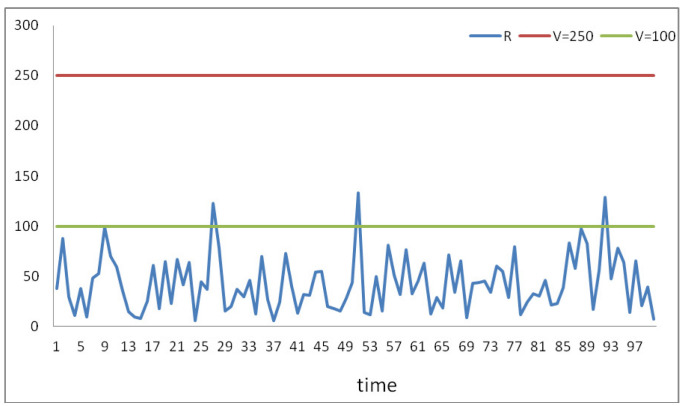
System simulation.

**Figure 8 sensors-22-02419-f008:**
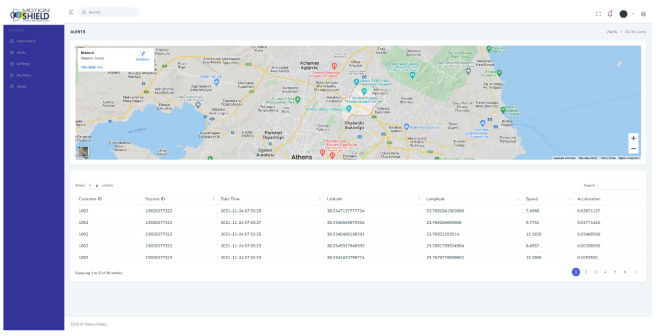
Motion Shield dashboard.

**Figure 9 sensors-22-02419-f009:**
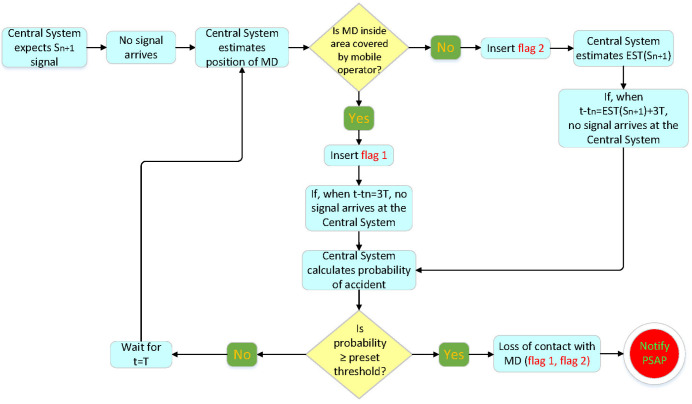
Description of what happens when the system loses contact with MD.

**Table 1 sensors-22-02419-t001:** Definitions.

MD	Mobile device (mobile phone, laptop, tablet, wearables).
Application	The end user application installed in an MD.
$S_{0}$	The first signal the application sends when it is activated and enters standby mode.
AS	Activation speed. The minimum moving speed of the mobile device that activates a new session and starts transmitting.
$S_{1}$	The signal transmitted when the application is in standby mode and the MD reaches a drive speed that exceeds the AS. Transmission of this signal indicates the start of a new session.
Session	The sequence of signals sent from the application, once the device acquires a speed higher than the AS. It begins with transmission of $S_{1}$ and ends with transmission of $S_{end}$.
T	The predetermined period (s) between two successive signals $S_{m}$ and $S_{m+1}\left(m\in N^{*}\right)$.
$S_{2},S_{3},\ldots ,S_{m}$	Defines the order of the regular predetermined signals sent successively after sending the $S_{1}$, during a session.
$S_{end}$	The signal sent by the application to notify the system that it terminates the session.
*u*	Mobile device moving speed.
$t_{n}$	The time elapsed between sending $S_{1}$ and $S_{n}$, (n≥2).
$ES_{1}$	Emergency signal related to the remaining battery charge of the MD, below a predetermined limit.
$ES_{2}$	Emergency signal related to user disconnection of the MD’s data transmission.
$ES_{3}$	Emergency signal related to user turning off their MD.
$ES_{4}$	Emergency signal indicating that the MD is probably involved in an accident.
$ES_{5}$	Emergency signal stating that the MD is subject to a vertical displacement exceeding a predetermined limit.
Data tran/sion	The application ability to send data, provided by either the mobile phone provider or wireless access of the MD.
*t*	The time elapsed between $S_{1}$ and now.
*t* _ *ES* _	The time elapsed since transmitting *ES*_4_, up to the current calculation time.
EST($S_{n}$)	After sending $S_{n}$, the estimated time needed for the MD to enter area with mobile coverage and send $S_{n+1}$. When the MD moves inside an area with mobile coverage, EST($S_{n}$) equals T. When it moves outside, EST($S_{n}$) is calculated dynamically, taking into account MD’s moving profile, road network topology, and current weather conditions.
SD	Safety distance between the MD and the geographical location from which $ES_{4}$ was sent. If after 3T+ST s from $ES_{4}$ transmission, MD’s distance is greater than SD, then the PSAP alert is canceled. SD is dynamically calculated based on the MD’s moving speed at the time $ES_{4}$ is sent, the road network topology, and the slippery condition of the road.
ST	Sliding time: after an accident, the estimated maximum time in s during which the MD is moving. ST is dynamically calculated based on the MD’s moving speed at the time $ES_{4}$ is sent, the road network topology, and the slippery condition of the road.
OBD-II	Onboard diagnostics protocol II.

**Table 2 sensors-22-02419-t002:** Visibility and speed parameters.

Visibility			Speed	
Range (m)		Weight	Km/h	Weight
2000	10,000	1	130	5
1000	1999	1.3	110	2.5
500	999	1.7	80	2
300	499	2	50	1.5
200	299	2.5	30	1.1
100	199	3.5	0	1
0	99	5		

**Table 3 sensors-22-02419-t003:** Road type and brightness parameters.

Road Network		Brightness	
Type	Weight	Time	Weight
National	1	Day (7 a.m.–7 p.m.)	1
Rural	1.2	Night (7 p.m.–7 a.m.)	1.5
Urban	1.5		

**Table 4 sensors-22-02419-t004:** Weather conditions and codes as they appear in Open Weather [9].

Weather Conditions		
ID	Phenomenon	Type
211	Thunderstorm	Thunderstorm
301	Drizzle	Drizzle
500	Rain	Light rain
521	Rain	Shower rain
601	Snow	Snow
615	Snow	Light rain and snow
701	Mist	Mist
741	Fog	Fog
800	Clear	Clear sky
803	Clouds	Broken: 51–84%
804	Clouds	Overcast: 85–100%

**Table 5 sensors-22-02419-t005:** Weather sub-type conditions.

Weather Sub-Type		
Weight: 1	Weight: 1.5	Weight: 2
200	201	202
210	230	212
211	231	221
300	302	232
301	312	313
310	502	314
311	511	321
500	521	503
501	531	504
520	612	522
600	616	602
601	620	603
615	731	613
701	751	621
721	761	622
711	803	741
800	804	762
801		771
802		781

## Data Availability

Not applicable.

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
