# Peer review of "Motion Shield: An Automatic Notifications System for Vehicular Communications"

_sensors, 2022, doi:10.3390/s22062419_

Round 1
Reviewer 1 Report
This paper presents an interesting apprach in which multiple information from the external enviroment are forwarded into a Decision Support System, which calculates dynamically the mobility risk and driving behavior aspects, in order to proactively send personalized notifications and alerts to the user and a Public Safety Answering Point (PSAP).
However, there are several shortcomings regarding the presentation of the issues covered in this article as well as the approval of its findings. Some of them are:
-The section "Background" should only contain the main terminology that the reader should know. An extra section "Literature Review" must be added in order an analytical comparison of the existing techniques with the main features and drawbacks of each technique or method compared with the proposed approach (maybe on a Table).
-The proposed architecture of the accident detection system is a layered architecture as is seen in Figure 1. Ηow data from from the smartphone sensors, location, and driver information is transferred to the cloud? The network layer utilises WiFi or 3G/4G cellular communications?
-How the Decision Support System that is presented estimate the likelihood of an accident involving a mobile device? There is not a relation for it in 4.2.3 subsection.
-For the verification of the proposed approach:
--How was the proposed approach evaluated, through simulations or through real experiments? A separate subsection for the experiment reuslts must be added. The evaluation setup must be explained.
--How was the data in Tables 2,3 produced? These values must be converted into realistic case studies.
--Additional metric evaluation must be added to evaluate the technique except of the severity such as the accuracy of accident
detection. How were the countemeasures proposed in section "3. Problem Description and Paper’s Contributions" evaluated in section "Experiment Results"?
Author Response
Please find attached the responses to the Reviewer's comments.

Reviewer 2 Report
A careful revision of the writing is required.
Numerous acronyms are not presented in the text.
Contributions must be carefully explained, please explain them. Give details.
Sometimes authors use numbers for bullets, other times points. Please, standardize.
Section 5 must be divided. In this sense, one section should be dedicated to RESULTS and discussion, this is the main drawback of the paper (Comparison results with other researches must be included by considering the most important metrics in the context). The final section should be conclusions and future works.
According to the style of the journal, abbreviations must be included at the end of the paper.
The references should be more recent, some of them are more than 5 years old.
Author Response

(The authors gave the same response as above.)

Round 2
Reviewer 1 Report
The idea presented in this paper is quite important (especially the proactively mobility risk assesment) and most of the reviewers' comments were answered by the authors. That is why I propose to publish it in current form. However, the idea must be sufficiently specialized both theoretically and practically with simulations because it still remains at a very theoretical stage.
Reviewer 2 Report
Accept in present form